# COVID-19 Pandemic Analysis for a Country's Ability to Control the Outbreak Using Little's Law: Infodemiology Approach

Yao-Hua Ho [1,*,†], Yun-Juo Tai [2,†] and Ling-Jyh Chen [1,2,3,*,†]

1   Department of Computer Science and Information Engineering, National Taiwan Normal University, Taipei 11677, Taiwan
2   Data Science Degree Program, College of Electrical Engineering and Computer Science, National Taiwan University and Academia Sinica, Taipei 10617, Taiwan; r08946010@ntu.edu.tw
3   Institute of Information Science, Academia Sinica, Taipei 11529, Taiwan
*   Correspondence: yho@ntnu.edu.tw (Y.-H.H.); ccllj@iis.sinica.edu.tw (L.-J.C.)
†   These authors contributed equally to this work.

**Abstract:** Since the outbreak of the coronavirus disease (COVID-19), all countries across the globe have been trying to control its spread. A country's ability to control the epidemic depends on how well its health system accommodates COVID-19 patients. This study aimed to assess the ability of different countries to contain the COVID-19 epidemic in real-time with the number of confirmed cases, deaths and recovered cases. Using the dataset provided by the Humanitarian Data Exchange (HDX), we analyzed the spread of the virus from 22 January 2020 to 15 September 2020 and used Little's Law to predict a country's ability to control the epidemic. According to the average recovery time curve changes, 16 countries are divided into different categories—Outbreak, Under Control, Second Wave of Outbreak, and Premature Lockdown Lift. The curves of outbreak countries (i.e., U.S., Spain, Netherlands, Serbia, France, Sweden, and Belgium) showed an upward trend representing that their medical systems have been overloaded and are unable to provide effective medical services to patients. On the other hand, after the pandemic-prevention policy was applied, the average recovery time dropped in under control countries (i.e., Iceland, New Zealand, Taiwan, Thailand, and Singapore). Finally, we study the impact of interventions on the average recovery time in some of the countries. The interventions, e.g., lockdown and gathering restrictions, show the effect after 14 days, which is the same as the incubation period of COVID-19. The results show that the average recovery time ($T$) can be used as an indicator of the ability to control the pandemic.

**Keywords:** infodemiological; coronavirus; COVID-19; outbreak control; Queuing Theory; Little's Law

## 1. Introduction

In December 2019, the novel coronavirus disease (COVID-19) appeared in Wuhan City and quickly became the epicenter of the epidemic. Before the world could realize the fatality of COVID-19 and the way it was transmitted, the virus had already spread to several nearby cities in the region. In less than two months, COVID-19 already surpassed the death toll of the severe acute respiratory syndrome (SARS) pandemic. Not only did the virus quickly spread to many neighboring countries, but also several cases were reported throughout Europe. To contain the virus and prevent outbreaks, it is essential to have accurate information to correctly estimate the spreading rate that helps in early detection and implementation of containment strategies. Taiwan is a leading example of how using accurate information and estimation regarding the virus based on early responses led to implementation of aggressive containment strategies, such as screening of passengers at airports, mandatory to wear a face mask, and a strict 14-days quarantine law with a smartphone location-aware monitoring program. Efficient strategies and accurate information provided Taiwan with a window of opportunity for early epidemic control. As a result of that, the country has reported only seven deaths with no community spread

of COVID-19 since the beginning of the epidemic. Without question, the Internet is the largest source for millions of people seeking information about the current status of COVID-19 around the world. The "Infodemiological" method, defined by Gunther Eysenbach [1], is the concept of information epidemiology that uses the Internet for user-contributed health-related content. The COVID-19 epidemiological data is compiled by the Johns Hopkins University Center for Systems Science and Engineering [2] from health departments in various countries. We obtained this data through the Humanitarian Data Exchange (HDX) database [3] and performed regional analysis to determine a country's ability to accommodate patients with COVID-19. In this context, we used Little's Law [4] to investigate and estimate how well a country's health system accommodates COVID-19 patients, which reflects their ability to control the epidemic.

## 2. Materials and Methods

### 2.1. Data Collection and Processing

We used the confirmed case data from the Humanitarian Data Exchange (HDX) for this study. HDX is managed by the Office for the Coordination of Humanitarian Affairs (OCHA), which is the part of the United Nations Secretariat responsible for bringing together humanitarian actors to ensure a coherent response to emergencies. Launched in July 2014, HDX is an open platform for sharing humanitarian data across crises and organizations which allow easy access and analysis. The collected datasets have been accessed by users in over 200 countries and territories. The dataset contains the daily cumulative number of confirmed, death and recovered cases from 22 January 2020 to 15 September 2020.

HDX is an open platform for collecting and sharing data across crises and organizations. HDX updates their data from different data contributors. The contributed data is checked by their Data Check that automatically detects errors with validation against CDCs and other vocabularies. However, if no data contributors update their current statistic data, then the data will be outdated, as in the case of the U.K. In addition, the data in the early period of the COVID-19 pandemic did not reflect on the ability of each countries' health care system to learn and adapt to manage pandemics. Our selection criteria of the countries were based on the availability of well documented data and information related to events and cases. Furthermore, we excluded the countries with outdated or unfitted data. For example, the U.K. did not update their number of recovered cases for a period of time. China's outbreak started on day one, which will not apply to our analysis.

### 2.2. Data Analysis

We analyze whether a country has a good control over the pandemic based on its daily cumulative number of confirmed cases, deaths and recovered cases. First, we start with an explanation of Queuing Theory. Queuing Theory is a simple mathematical study to predict lengths and waiting times of queues or waiting lines. It is often used in different areas from engineering to business management to determine the resources needed to provide a service.

One of the simplest queuing models is Little's Law [4]. Little's Law was proposed by John Little in 1961, and it states that the average number $N$ of customers in a stationary system is equal to the average arrival rate $\lambda$ multiplied by the average time $T$ that a customer spends in the system. The main advantage of Little's Law is to provide an intuitive approach for the assessment of the efficiency of a queuing system, in our case a country's health system. By applying the law, it is possible to get a better understanding of the capacity of each country for handling COVID-19 pandemic. By using up-to-date data (i.e., the total number of the confirmed cases, the death cases, and the recovered cases) from HDX, the average recovery time ($T$) estimated from Little's Law can be used as an indicator of the ability on controlling the pandemic. Algebraic expression of the theorem is

$$N = \lambda \times T, \tag{1}$$

For example, in a small bank with a single teller, where only one customer can be at the teller at a time, if an average of 20 customers arrive every hour (i.e., $\lambda = 20$ per h) and stay in the bank for 30 min (i.e., $T = 0.5$ h), then the average number of customers in the bank will be 10 customers (i.e., $N = 10$ customers) at any given time.

The system is in a stable state if the arrival rate $\lambda$ (e.g., customers arriving) to the system is less or equal to the rate of exiting the system. On the other hand, if the arrival rate is greater than the exit rate, the system goes in an unstable state.

Applying Little's Law to COVID-19, $N$ represents the prevalence of COVID-19, which is the proportion of infected people in a given time period, i.e., the total number of the confirmed cases minus the death cases and the recovered cases. The given time period is the window size $W$. In our study, the window size is twice the maximum length of the COVID-19 cases, which is about 2 months (60 days), i.e., 120 days. The average arrival rate $\lambda$ is the incidence rate per day and $T$ represents the average recovery time (i.e., the time from onset to clinical recovery or death) of confirmed cases.

Little's Law holds when the pandemic is under good control, i.e., in a stable state. In such cases, $T$ is expected to be stable in a reasonable range, and the value of $N$ may vary with $\lambda$ up to a certain value that represents the capacity of a health system.

When an outbreak occurs, the Little's Law no longer holds, and $\lambda$ may increase sharply, resulting in an extensive number of active confirmed cases ($N$) and out of its health system capacity. During that period, the value of $T$ also increases due to limited healthcare resources, and it may become even greater than the reasonable range.

In order to recover from the outbreak, it is essential to lower $N$ under its health system capacity. There are two ways to put a country back in good control of the pandemic. First, it needs to reduce $\lambda$ by promoting mask-wearing, strengthening quarantine measures, and tightening border control. Second, a country needs to reduce $T$ by improving treatments (i.e., increasing recovery rate) and patient prioritization.

## 3. Results

We marked the time points of the policies and the disease outbreak of multiple countries to observe whether these countries are in control of the epidemic. According to the changes in the average recovery time curve, we divide the countries into different categories: (a) Outbreak—U.S., Spain, Netherlands, Serbia, France, Sweden, and Belgium; (b) Under Control—Iceland, New Zealand, Taiwan, Thailand, and Singapore; (c) Second Wave of Outbreak—Australia and Hong Kong; and (d) Premature Lockdown Lift—Malta and Spain.

The curve of countries that have lost control of COVID-19 will show an upward trend, see Figure 1. The medical systems in these countries have been overloaded and are unable to provide effective medical services to patients. Therefore, as the number of confirmed cases increases, patients will stay in the disease state for a longer time. In other words, the overall average recovery time will be longer.

### 3.1. Outbreak Cases

First, we study some of known countries that are out of control such as the U.S., Spain, Netherlands, Serbia, France, Sweden, and Belgium. In Figure 1, all seven countries are experiencing a rapid increase of newly confirmed cases (i.e., average arrival rate $\lambda$). As the $\lambda$ increased, the number of active confirmed cases ($N$) also increased sharply, which most likely exceeded each countries' health system capacity. The steeper slope means rapid growth in newly confirmed cases in a country. This implies an increase in the average recovery time ($T$) to an unreasonable range when an outbreak occurs.

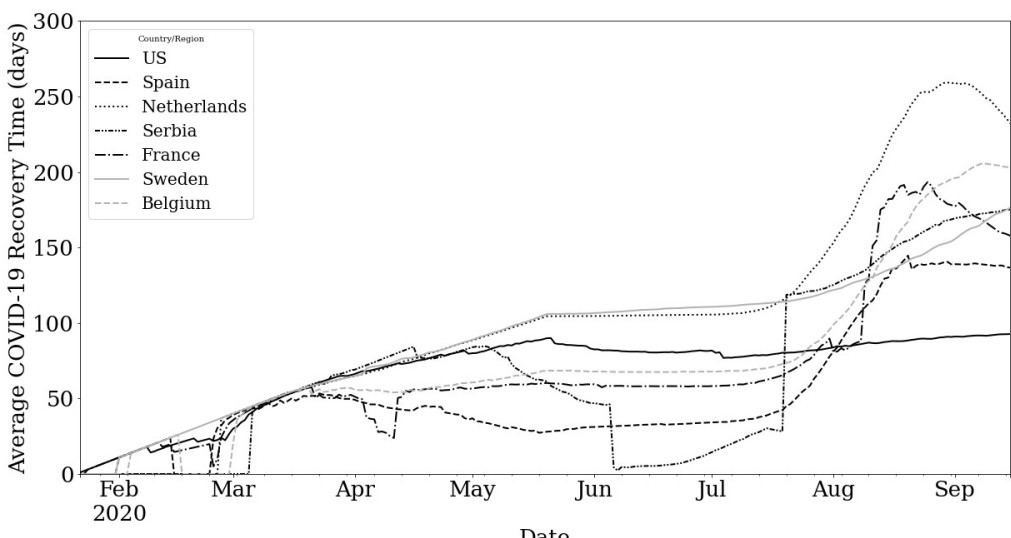

**Figure 1.** Average Recovery Time for U.S., Spain, Netherlands, Serbia, France, Sweden and Belgium.

*3.2. Under Control Cases*

In this section, we study countries with better control of the COVID-19 pandemic. We defined a country as well under control if its number of new confirmed cases ($N$) is less than 7 cases per day for 30 consecutive days and the average recovery time ($T$) is less than 20 days. The five well under control countries are Iceland, New Zealand, Taiwan, Thailand, and Singapore.

In Figure 2, all four countries adopted policies such as the lockdown or border control during the first wave of the epidemic, and only lifted restrictions after the epidemic was well controlled. Iceland banned gathering and implemented travel restrictions in mid-March. Afterward, Iceland enforced stricter gathering restrictions on 24 March 2020 (I1) [5] and partially eased the lockdown after 4 May 2020 (i.e., I2) [6]. When the number of new domestic cases started to increase around 23 July 2020 (I3) [7], Iceland tightened the restrictions on 30 July 2020 (I4) [8]. New Zealand started their lockdown on 25 March 2020 (N1) [9] and partially eased the lockdown after 14 May 2020 (N2) [10]. When familial cluster infection cases appeared in Auckland on 11 August 2020 (N3) [11], New Zealand reinforced the lockdown rules in Auckland on 14 August 2020 (N4) [12] after the cluster outbreak. Comparing to the countries with outbreak cases, Iceland and New Zealand have the advantage of geographic isolation for border control during the first wave of the COVID-19 pandemic.

With the previous experiences with SARS outbreaks in 2014, Taiwan was very cautious and paid close attention to this new respiratory disease due to the similarity between COVID-19 and SARS. Thus, Taiwan took immediate actions after the number of new imported cases began increasing from 15 March 2020 (TW1) [13], such as prohibiting foreign nationals from entering their country on 19 March 2020 (TW2) [14] and made it mandatory to wear a mask on public transportation on 1 April 2020 (TW3) [15]. They started to lift the restrictions, such as the limitation of sport game spectators, from 6 May 2020 (TW4) [16]. Taiwan started facing the second wave of the epidemic on 27 July 2020 (TW5) [17]. Since Taiwan still required a 14-day quarantine for people entering the country, the epidemic was still under control, and the average recovery time fluctuated rather than increased steadily.

Thailand announced a cluster on 12 March 2020 (TH1) [18] and the outbreak of infection cases in the boxing stadium on 15 March 2020 [19]. Subsequently, the Thailand government declared a state of emergency from 26 March 2020 (TH2) [20]. Related measures included lockdown and the closure of borders. Thailand started four stages of relaxation of restrictions from 3 May 2020 (TH3) [21] and eased the entry restrictions on 1 July 2020 (TH4) [22]. Afterward, Thailand closed the border again to prevent an increase in the

number of imported cases on 14 July 2020 (TH5) [23]. Thailand is one of the ten members of ASEAN countries. Although, Thailand has lower GDP compared with many countries; we compared Thailand due to its popularity as a vacation destination for foreigners. According to the results, the country regained control of the COVID-19 pandemic after enforcing the policy of closing its border.

Similarly, Singapore is one of the ten members of ASEAN countries with a total land area of only 724.2 square kilometers. We compared Singapore as it is one of the top financial centers having a large number of international companies with a large number of foreign workers working in limited office spaces. Singapore experienced the outbreak after The Life Church and Missions Singapore outbreak in February of 2020 (i.e., SG1, 29 January 2020) and the SAFRA Jurong cluster in March (i.e., SG2, 27 February 2020) [24]. After the outbreak, Singapore started enforcing the lockdown rules step-by-step with different rules, such as closing border on 24 March 2020 (i.e., SG3) [25], implementing partial lockdown on 3 April 2020 (i.e., SG4, circuit breaker measures, which advised to stay home as much as possible) [26], enforcing mask mandates on 14 April 2020 [27], tightening measures of circuit breaker on 21 April 2020 (i.e., SG5) [28] and scaling-up testing on 27 April 2020 [29]. In Singapore, the government started to reopen activities from May 2020 slowly. The lockdown was lifted on 2 May 2020 [30] and the entry restrictions were eased on 17 June 2020 (i.e., SG6) [31]. Meanwhile, the government still expanded the coverage of testing to reduce the risk of cluster infections [32]. Due to the early detection, the number of new confirmed cases, i.e., the arrival rate $\lambda$, decreased, which led to a decrease in the average recovery time.

All five countries (Iceland, New Zealand, Taiwan, Thailand, and Singapore) waited until the COVID-19 pandemic was well under control before they eased their lockdown restrictions. As a result, those countries were able to keep the number of confirmed cases ($N$) below their health system capacity, which lowered the average arrival rate ($\lambda$) and kept the countries in a stable state. Little's Law holds when the pandemic is under control.

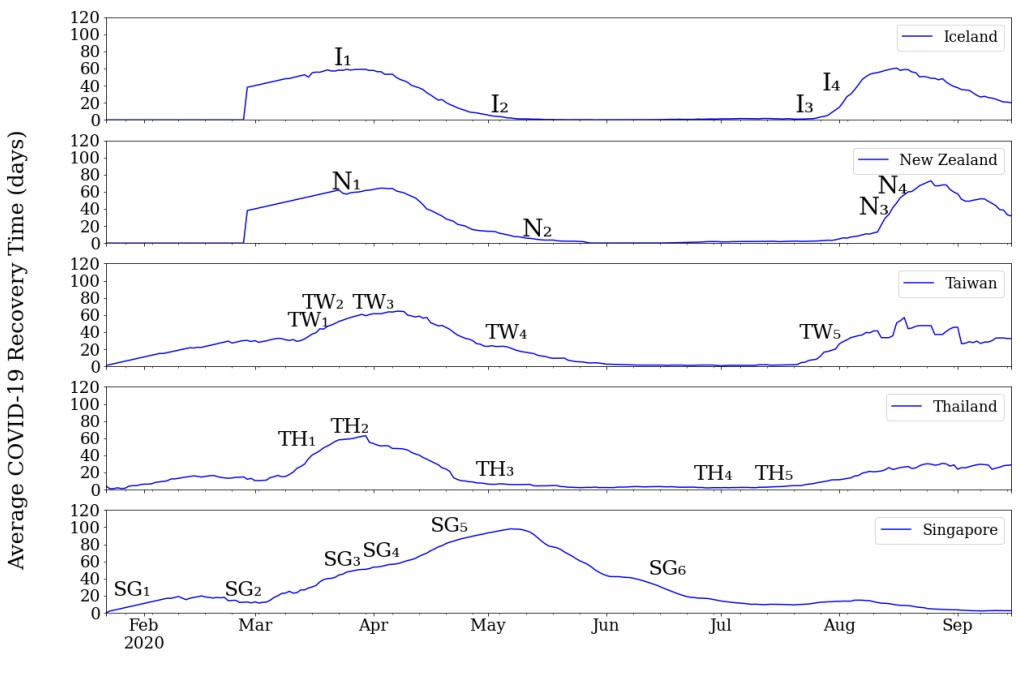

**Figure 2.** The average recovery time for COVID-19 of Under Control Cases with time points of the policies and the disease outbreak. From top to bottom are Iceland, New Zealand, Taiwan, Thailand and Singapore.

### 3.3. Second Wave of Outbreak Cases

In this section, we study Australia and Hong Kong that have experienced the 2nd wave of outbreak. In both countries, the lockdown rules started after the first wave of the outbreak. In Figure 3, Australia and Hong Kong began their measures for social distancing on 18 March 2020 (A1) [33] and 29 March 2020 (H1) [34], respectively.

After the lockdown rules, the number of newly confirmed cases (i.e., average arrival rate $\lambda$) started to decrease in both countries. However, the second wave of the outbreak started to appear as nations loosen their lockdown rules. In Australia, a significant community transmission was found among 11 new cases on 23 June 2020. [35]. Then, the local outbreaks from 6 July 2020 (A2) [36] led to a surge in the number of new cases and resulted in the second wave of the outbreak. Australia was able to regain control by reinforcing the lockdown rules across the state of Victoria, including metropolitan Melbourne (A3) [37]. Similarly, the second wave situation began in Hong Kong with a community transmission on 2 June 2020 [38]. Then, cases of local infections were reported in multiple districts on 6 July 2020 (H2 and H3) [39].

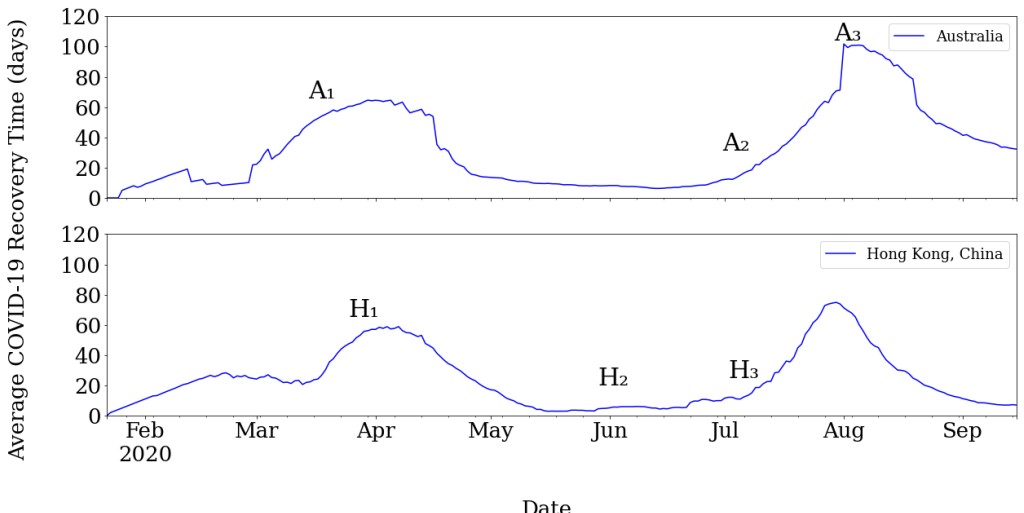

**Figure 3.** The average recovery time for COVID-19 of Second Wave of Outbreak Cases with time points of the policies and the disease outbreak. From top to bottom are Australia and Hong Kong (China).

### 3.4. Premature Lift of Lockdown Cases

In this section, we study two countries, Malta and Spain, which lifted the lockdown too early. In Figure 4, Malta started the lockdown restriction on 22 March 2020 (M1) [40]. Shortly, it eased the lockdown on 4 May 2020 (M2) [41] when the average recovery time ($T$) is still about 20 days. Then, the country lifted restrictions for all arrival planes on 15 July 2020 (M3) [42] when $T$ is about 1. However, the mass events, e.g., Catholic Saint feast celebrations and Feast of St. Venera, at the end of July 2020 (M4) [43] caused the second wave in Malta.

Similarly, Spain started the national lockdown on 15 March 2020 (SP1) [44]. Soon, it started to lift the restrictions on 2 May 2020 (SP2) [45] and then implemented the first and second stage of easing lockdown on 11 May 2020 and 18 May 2020, respectively. Furthermore, it ended the national state of alarm on 21 June 2020 (SP3) [46]. However, the cluster infection appeared in Segria (SP4) [47] and there was an outbreak at the local bars in Totana (SP5) [48]. Compared to Taiwan and New Zealand, the ease is too early to ensure the outbreak has been completely controlled. As a result of prematurely lifting their lockdowns, the second wave outbreak appeared in both countries at the end of July.

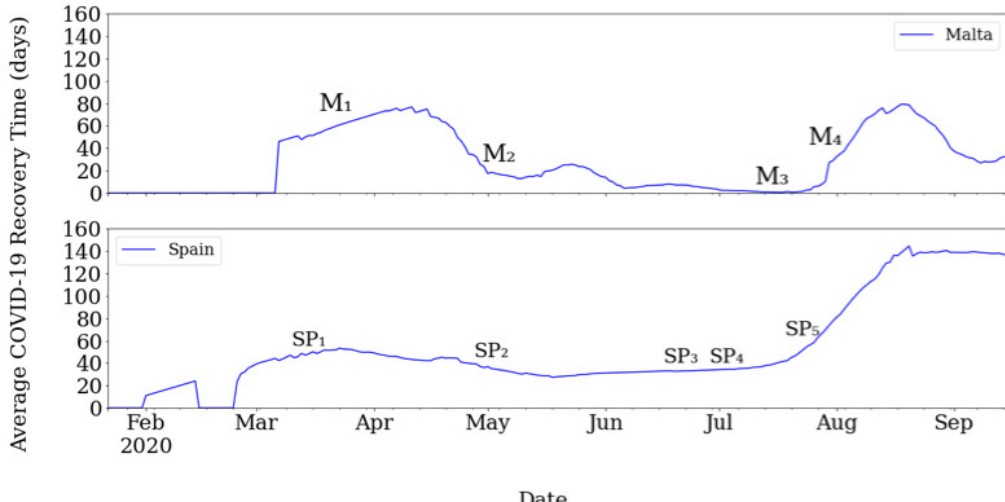

**Figure 4.** The average recovery time for COVID-19 of premature lift of lockdown cases with time points of the policies and the disease outbreak. From top to bottom are Malta and Spain.

## 4. Discussion

The result shows that the average recovery time can reflect the severity of the pandemic. When a cluster infection occurred, the average recovery time increased rapidly with the number of new cases. After the pandemic-prevention policy was applied, the average recovery time dropped as the number of new confirmed cases decreased. We also observed that the prevention policy, e.g., lockdown and gathering restrictions, shows the effect after 14 days, which is the same as the incubation period of COVID-19.

There are some advantages of using the average recovery time as a pandemic severity index. First, the average recovery time requires fewer parameters to be estimated. Compared with $R_0$, which is usually needed to build an epidemic model with many hyperparameters, the average recovery time only needs to consider the window size. Second, the average recovery time takes the influence of health system capacity into account. The higher the health system capacity of a country, the higher the ability to control the epidemic.

The window size ($W$) will affect the sensitivity of the index, i.e., the average recovery time. A large window size flattens the curve of the average recovery time and ignores the occurrence of small cluster infections. In contrast, a small window size overestimates the average recovery time and fails to highlight the outbreak. According to the Sampling Theorem, the sampling frequency should be twice the sampling rate. The COVID-19 report from WHO shows that the clinical recovery time of COVID-19 patients is about 2–6 weeks. Furthermore, the time for the onset of the disease to death is about 2–8 weeks [49]. Therefore, we use twice the maximum length of the average COVID-19 cases, which is about 2 months, as the window size.

## 5. Conclusions

In this paper, we use Little's Law to estimate the hospital capacity for the COVID-19 pandemic of each country and find out a new estimator to represent the severity of the pandemic. We study the impact of interventions on the average recovery time in some countries. The results show that the average recovery time ($T$) can be used as an indicator of the ability to control the pandemic. In the near future, we plan to study the scaling of window size and forecasting of the average recovery time for pandemic prevention.

**Author Contributions:** Conceptualization, Y.-H.H. and L.-J.C.; methodology, L.-J.C.; software, Y.-J.T.; validation, Y.-H.H., Y.-J.T. and L.-J.C.; formal analysis, Y.-J.T.; investigation, Y.-J.T.; resources, Y.-H.H. and L.-J.C.; data curation, Y.-J.T.; writing—original draft preparation, Y.-H.H.; writing—review and editing, Y.-J.T. and L.-J.C.; visualization, Y.-H.H. and Y.-J.T.; supervision, Y.-H.H. and L.-J.C.; project administration, L.-J.C. All authors have read and agreed to the published version of the manuscript.

**Funding:** This research was funded by Ministry of Science and Technology of Taiwan under Grant No. MOST 109-2218-E-001-002.

**Institutional Review Board Statement:** Not applicable.

**Informed Consent Statement:** Not applicable.

**Data Availability Statement:** Publicly available datasets were analyzed in this study. This data can be found here: [https://data.humdata.org/dataset/novel-coronavirus-2019-ncov-cases] (accessed on 30 September 2020).

**Conflicts of Interest:** The authors declare no conflict of interest.

## Abbreviations

The following abbreviations are used in this manuscript:

| | |
|---|---|
| COVID-19 | Coronavirus Disease 2019 |
| HDX | Humanitarian Data Exchange |
| OCHA | Office for the Coordination of Humanitarian Affairs |
| SARS | Severe Acute Respiratory Syndrome |
| MDPI | Multidisciplinary Digital Publishing Institute |
| DOAJ | Directory of open access journals |

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
