# Peer review of "COVID-19 Pandemic Analysis for a Country’s Ability to Control the Outbreak Using Little’s Law: Infodemiology Approach"

_sustainability, doi:10.3390/su13105628_

Round 1

Reviewer 1 Report

The paper by Yao-Hua Ho ingeniously analyzes the critical issues of patient streamlining for the diagnosis and treatment of COVID 19. This using 4 "frames" of national epidemic behavior. The assumptions on which it is based is Little law and derive from the data available from  HDX database. According to the general principles we can be confident that HDX database are good quality sources. There remain incertitudes about some governmental sources that may have underestimated and minimized the volumes of infections (think to authorities close to the Olympics deadlines). The availability of molecular testing for nasopharyngeal swab was another confounder together with COVID-19 variants that may depict different diffusibility patterns.

Remarks

1-Please refine the discussion adding that countries in "under control cases" are insular countries with peculiar conditions. Iceland and New Zealand may  be considered as remote areas of the planet with excellent chances of isolation. Taiwan and Singapore experienced previous infectious alarms 2004-2015 with efficient preparedness since SARS outbreaks. ASEAN countries  can hardly be compared with other nations for GDP and social aspects to Taiwan or Iceland.

3-Again improve the discussion stating the limits of the study in the discussion: HDX reports may be at risk of omissions or overestimation for different availability of diagnostic tools (SARS-prepared countries).  A learning curve for laboratories and diagnostic facilities was observed for many health care systems not organized to manage pandemics. 

2-The legends may be improved adding a sentence like: "time points for average recovery time for COVID-19 ...."

3-The authors need to state data sources in the HDX database. Reference 1 shall be in line 40. While a direct reference to HDX database shall be in line 41.

4-Line 178 "appeared" check tenses I think is wrong

5-Line 179 check sentence the meaning is not clear

Author Response

Responses to Reviewers’ Comments

Yao-Hua Ho, Yun-Juo Tai, and Ling-Jyh Chen

RE: COVID-19 Pandemic Analysis for a Country’s Ability to Outbreak Control using Little’s Law: Infodemiology Approach

Author's Reply to the Review Report

 Reviewer 1 - Comments and Suggestions for Authors

The paper by Yao-Hua Ho ingeniously analyzes the critical issues of patient streamlining for the diagnosis and treatment of COVID 19. This using 4 "frames" of national epidemic behavior. The assumptions on which it is based is Little law and derive from the data available from  HDX database. According to the general principles we can be confident that HDX database are good quality sources. There remain incertitudes about some governmental sources that may have underestimated and minimized the volumes of infections (think to authorities close to the Olympics deadlines). The availability of molecular testing for nasopharyngeal swab was another confounder together with COVID-19 variants that may depict different diffusibility patterns.

RESPONSE: We wish to thank the reviewer for all the valuable comments and suggestions, which have helped us improve the quality of our manuscript. We agree with the reviewer. The issue of underestimation of the volumes of infection can negatively impact the capacity of each country for handling COVID-19 pandemic. This can lead to incorrect decisions and delay in implementation of containment strategies by the authorities. The availability of molecular testing and different COVID-19 vaccines will definitely result different diffusibility patterns. Our future research will include development of models that would consider complex issues related to testing practices, COVID-19 variants and underestimation of cases in certain countries.

To address, the reviewer additional remarks, our responses are listed below and attached file for our updated paper.

Remarks:

  • Please refine the discussion adding that countries in "under control cases" are insular countries with peculiar conditions. Iceland and New Zealand may  be considered as remote areas of the planet with excellent chances of isolation. Taiwan and Singapore experienced previous infectious alarms 2004-2015 with efficient preparedness since SARS outbreaks. ASEAN countries  can hardly be compared with other nations for GDP and social aspects to Taiwan or Iceland.

RESPONSE: We thank the reviewer for valuable comments. The reviewer correctly points out that countries in “under control cases” are with peculiar conditions, e.g., geographic isolation and previous experiences with SARS outbreaks. Although, Thailand and Singapore are two of the ten members of ASEAN countries. We compared Thailand due to its popularity of vacation destination for foreigners. Singapore is one of the top financial centers with large number of foreign companies and foreign workers in ASEAN countries. After the policies of the closure of borders in two countries, the results showed both countries can regain the control of the COVID-19 pandemic.

We have added the discussion (in blue text) regarding the countries with geographic isolation and previous experiences with SARS outbreaks in Section 3.2.

“Comparing to the countries with outbreak cases, Iceland and New Zealand have the advantage of geographic isolation for border control during the first wave of the COVID-19 pandemic.”

“With the previous experiences with SARS outbreaks in 2014, Taiwan was very cautious and paid close attention to this new respiratory disease due to the similarity between COVID-19 and SARS. Thus, Taiwan took immediate actions after the number of new imported cases start increasing from 03/15/2020 (TW1) ”

“Thailand is one of the ten members of ASEAN countries. Although, Thailand has lower GDP compare with many countries; we compared Thailand due to its popularity as a vacation destination for foreigners. According to the results, the country regained control of the COVID-19 pandemic after enforcing the policy of closing its border.”

“Similarly, Singapore is one of the ten members of ASEAN countries with a total land area of only 724.2 square kilometers. We compared Singapore as it is one of the top financial centers having a large number of international companies with a large number of foreign workers working in limited office spaces.”

  • Again improve the discussion stating the limits of the study in the discussion: HDX reports may be at risk of omissions or overestimation for different availability of diagnostic tools (SARS-prepared countries).  A learning curve for laboratories and diagnostic facilities was observed for many health care systems not organized to manage pandemics. 

RESPONSE: We thank the reviewer for valuable comments. The reviewer accurately points out the issues and the limitation of HDX data for some countries. In Section 2.1, we have discussed the countries we selected are based on the well-known data, events, and cases. Also, we excluded the countries with outdated or unfitted data, i.e., U.K. and China.

In addition to the added discussions in the above reply to reviewer’s remark, we added more discussion (in blue text) in the Section 2.1.

“HDX is an open platform for collecting and sharing data across crises and organizations. HDX update their data from different Data Contributors. The contributed data will be checked by their Data Check that automatically detects errors with validation against CDCs and other vocabularies. However, if no data contributors update their current statistic data, then the data will be out dated, as in the case of U.K. In addition, the data in the early period of the COVID-19 pandemic did not reflect on the ability of each countries’ health care system to learn and adapt on manage pandemics.”

  • The legends may be improved adding a sentence like: "time points for average recovery time for COVID-19 ...."

RESPONSE: We thank the reviewer for the valuable comment. We have sent the paper to a native English speaker for English proofreading and editing to address the writing issue.

  • The authors need to state data sources in the HDX database. Reference 1 shall be in line 40. While a direct reference to HDX database shall be in line 41.

RESPONSE: We thank the reviewer for  pointing out the missed references. We have updated the paper by adding the data source.

” The COVID-19 epidemiological data is compiled by the Johns Hopkins University Center for Systems Science and Engineering [1] from health department in various countries. We obtained this data through the Humanitarian Data Exchange (HDX) database [2] and performed regional analysis to determine a country’s ability to accommodate patients of COVID-19.”

(Added two references)

  1. Dong, E., Du, H., Gardner, L. An interactive web-based dashboard to track COVID-19 in real time. The Lancet infectious diseases 2020, 20(5), 533--534.
  2. Humanitarian Data Exchange (HDX). Available online: https://data.humdata.org (accessed on 30 Sept 2020).

  • Line 178 "appeared" check tenses I think is wrong

RESPONSE: We thank the reviewer for pointing it out. We have sent the paper to a native English speaker for English proofreading and editing to address the writing issue.

  • Line 179 check sentence the meaning is not clear

RESPONSE: We thank the reviewer for pointing it out. We have sent the paper to a native English speaker for English proofreading and editing to address the writing issue.

Reviewer 2 Report

This is more of a re-presentation of the results than a discussion. A very current topic. Very professionally guided reasoning and processing of results.  Discussion is to short. This is more of a re-presentation of the results than a discussion. It doesn't really show the advantages of Little's Law.

Author Response

Responses to Reviewers’ Comments

Yao-Hua Ho, Yun-Juo Tai, and Ling-Jyh Chen

RE: COVID-19 Pandemic Analysis for a Country’s Ability to Outbreak Control using Little’s Law: Infodemiology Approach

Author's Reply to the Review Report

Reviewer 2 - Comments and Suggestions for Authors

This is more of a re-presentation of the results than a discussion. A very current topic. Very professionally guided reasoning and processing of results.  Discussion is too short. This is more of a re-presentation of the results than a discussion. It doesn't really show the advantages of Little's Law.

RESPONSE: We wish to thank the reviewer for all the valuable comments and suggestions, which have helped us improve the quality of our manuscript. The main contribution of the paper is to apply open access data, such as HDX, to estimate how well a country’s health system accommodates COVID-19 patients and how it reflects on their ability to control the epidemic using a simplest queuing model, Little’s Law.

We have added the description and explanation (in blue text) to discuss the advantage of Little’s Law in Section 2.2.

“The main advantage of Little’s Law is to provide an intuitive approach for the assessment of the efficiency of a queuing system, in our case a country’s health system. By applying the law, it is possible to give us a better understanding on the capacity of each country for the COVID-19 pandemic. By using up-to-date data (i.e., the total number of the confirmed cases, the death cases, and the recovered cases) from HDX, the average recovery time (T) estimated from Little’s Law can be used as an indicator of the ability on controlling the pandemic.”

Reviewer 3 Report

Must improve English, and be edited by a native English speaker.

Author Response

Responses to Reviewers’ Comments

Yao-Hua Ho, Yun-Juo Tai, and Ling-Jyh Chen

RE: COVID-19 Pandemic Analysis for a Country’s Ability to Outbreak Control using Little’s Law: Infodemiology Approach

Author's Reply to the Review Report

Reviewer 3 - Comments and Suggestions for Authors

Must improve English, and be edited by a native English speaker.

RESPONSE: We wish to thank the reviewer for the valuable suggestion, which have helped us improve the quality of our manuscript. We have sent the paper to a native English speaker for English proofreading and editing to address the writing issue. Please see the attached file for our updated paper.

Round 2

Reviewer 1 Report

Point-by-point Response to the reviewers' comments are appropriate.

Author Response

Once again, we wish to thank the reviewer for all the valuable comments and suggestions, which have helped us improve the quality of our manuscript. 

Reviewer 3 Report

Sustainability_review                                                                                      May 7, 2021

The article “Covid-19 Pandemic Analysis for a Country’s Ability to Outbreak Control using Little’s Law:  Infodemiology Approach”, is an interesting article.  The article however, needs additional editing:

Working from the currently revised article:

Title:  perhaps “Control the Outbreak” reads better.

Line 27:  “Not only did the virus…”

Line 38:  “mandatory to wear a face mask, and…”

Line 48:  “patients with Covid-19”.

Line 50:  “patients which reflects their ability…”

Line 74:  “Day One”, why are these words capitalized?

Line 97:  come vs. arriving? Use of the word come is a bit odd sounding here.

Line 98:  remove of.

Line 107:  remove the.

Line 111:  “confirmed cases (N) and out of…”

Line 113:  remove far.

Line 117, 118:  avoid use of ect.  List that which you want to state.

Line 121:  remove well.

Figure 1:  the legend is impossible to read in black and white, is there a way to modify the (dashed vs. solid), so that legible in printed format.

Lines 143, 145:  should read well under control.

Regarding tenses, it is not clear where the use of present and past should be used.  Please clarify, sections 3.2, and 3.3.

Line 192:  what is a circuit breaker?

Paragraphs on Singapore, should these be joined into one?  Is there a reason these are separate.

Line 208:  remove good.

Line 228:  3.4 “Premature Lockdown-Lift Cases”, this reads a bit awkward.  Perhaps could read:  Premature Lift of Lockdown

Line 268-269:  Use for what, this is not clear?

Author Response

Once again, we wish to thank the reviewer for all the valuable comments and suggestions, which have helped us improve the quality of our manuscript.

We have updated the paper by addressing the writing issues pointed out by the reviewer (in the blue text of the track changes version of the paper). 

Round 3

Reviewer 3 Report

Sustainability_review                                                                                      May 7, 2021

The article “Covid-19 Pandemic Analysis for a Country’s Ability to Outbreak Control using Little’s Law:  Infodemiology Approach”, is an interesting article.  The article however, needs additional editing:

Working from the currently revised article:

Title:  perhaps “Control the Outbreak” reads better.

Line 27:  “Not only did the virus…”

Line 38:  “mandatory to wear a face mask, and…”

Line 48:  “patients with Covid-19”.

Line 50:  “patients which reflects their ability…”

Line 74:  “Day One”, why are these words capitalized?

Line 97:  come vs. arriving? Use of the word come is a bit odd sounding here.

Line 98:  remove of.

Line 107:  remove the.

Line 111:  “confirmed cases (N) and out of…”

Line 113:  remove far.

Line 117, 118:  avoid use of ect.  List that which you want to state.

Line 121:  remove well.

Figure 1:  the legend is impossible to read in black and white, is there a way to modify the (dashed vs. solid), so that legible in printed format.

Lines 143, 145:  should read well under control.

Regarding tenses, it is not clear where the use of present and past should be used.  Please clarify, sections 3.2, and 3.3.

Line 192:  what is a circuit breaker?

Paragraphs on Singapore, should these be joined into one?  Is there a reason these are separate.

Line 208:  remove good.

Line 228:  3.4 “Premature Lockdown-Lift Cases”, this reads a bit awkward.  Perhaps could read:  Premature Lift of Lockdown

Line 268-269:  Use for what, this is not clear?

These edits have not been addressed.

Author Response

Once again, we wish to thank the reviewer for the valuable suggestion, which have helped us improve the quality of our manuscript.

We have corrected the errors pointed out by the reviewer (in blue text). Please see attached paper. 
